# Integration of InSAR and LiDAR Technologies for a Detailed Urban Subsidence and Hazard Assessment in Shenzhen, China

**Yufang He [1,2], Guochang Xu [1,*], Hermann Kaufmann [3], Jingtao Wang [2], Hua Ma [4] and Tong Liu [1]**

1. Institute of Space Science and Applied Technology, Harbin Institute of Technology (Shenzhen), Shenzhen 518055, China; heyufang0823@163.com (Y.H.); liutong2017@126.com (T.L.)
2. Shenzhen Municipal Planning, & Land Realestate Information Center (Shenzhen GeoSpatial Information Center), Shenzhen 518034, China; william.jt.wang@connect.polyu.hk
3. School of Space Science and Physics, Shandong University at Weihai, Weihai 264209, China; Hermann-kaufmann@t-online.de
4. Shenzhen Lijian Skyeye-Laser Technology Limited Company, Shenzhen 518055, China; suiyuan009@163.com
* Correspondence: xuguochang@hit.edu.cn; Tel.: +86-0755-26611417

**Abstract:** Spaceborne interferometric synthetic aperture radar (InSAR) methodology has been widely successfully applied to measure urban surface micro slow subsidence. However, the accuracy is still limited by the spatial resolution of currently operating SAR systems and the lacking precision of geolocation of the respective scatters. In this context, high-precision urban models, as provided by the active laser point cloud methodology through light detection and ranging (LiDAR) techniques, can assist in improving the geolocation quality of InSAR-derived permanent scatters (PS) and provide the precise contour of buildings for hazard analysis. This paper proposes to integrate InSAR and LiDAR technologies for an improved detailed analysis of subsidence levels and a hazard assessment for buildings in the urban environment. By the use of LiDAR data, most building contours in the main subsidence area were extracted and SAR positioning of buildings via PS points was refined more precisely. The workflow for the proposed method includes the monitoring of land subsidence by the TS-InSAR technique, the geolocation improvement of InSAR-derived PS, and building contour extraction by LiDAR data. Furthermore, a reasonable hazard assessment system of land subsidence was developed. Significant vertical subsidence of −40 to 12 mm per year was detected by the analysis of multisensor SAR images. The land subsidence rates in the Shenzhen District obviously follow certain spatial patterns. Most stable areas are located in the middle and northeast of Shenzhen except for some areas in Houhai, the Qianhai Bay, and the Wankeyuncheng. An additional hazard assessment of land subsidence reveals that the subsidence of buildings is mainly caused by the construction of new buildings and some by underground activities. The research results of this paper can provide a useful synoptic reference for urban planning and help reducing land subsidence in Shenzhen.

**Keywords:** urban subsidence; hazard assessment; reclamation areas; InSAR; LiDAR

## 1. Introduction

The spaceborne InSAR technology comprises the advantages of monitoring large areas almost contemporarily, with an all-weather, all-day capability and a high degree of automatic processing, and can be used for accurate measurements of surface deformations such as urban land subsidence [1–3], landslide monitoring [4,5], earthquake analysis [6], infrastructure assessment [7,8], etc. In this context, urban land subsidence has recently been recognized as an ongoing serious process causing damages to urban buildings and danger to the population. Therefore, the monitoring and hazard assessment of urban buildings in growing cities are of particular importance. Using a contemporary development in InSAR applications, the so-called time series interferometric synthetic aperture radar (TS-InSAR) method, atmospheric effects can be estimated [9–13]. Thus, the accuracy of digital elevation

models (DSMs) is optimized for a precise monitoring of micro slow subsidence rates, and the potential of TS-InSAR for the detection of dislocations of urban buildings is enhanced. However, coherent radar scatters are often characterized by a rather poor geopositioning with an accuracy of 3–4 pixels (even more pixels for buildings), and urban surface objects including buildings, roads and bridges are difficult to separate and identify [14]. Therefore, it is hardly feasible to correlate accurately coherent radar scatters such as those mentioned above and those derived by the TS-InSAR method with real ground targets. To overcome this problem, airborne LiDAR technology is used, which can obtain the three-dimensional spatial information of ground objects directly, and has the advantages of only minor impacts from the climate, high positioning accuracy, and short production cycles [15]. The high-precision city models based on this technology can provide external geometric contours and very high accuracy of positioning information for urban buildings [16,17]. Subsequently, due to the DSM data derived from the LiDAR technique, the computed InSAR PS geolocations can be much better refined. In recent years, the precision of PS geopositioning of high-resolution SAR data was studied by a number of researchers [18,19]. A passive angle repeater (CR) was used to improve the accuracy of the X-band (TerraSAR-X) PS geolocation, which reduces the uncertainty of a geographic location to a subcentimeter level [20]. LiDAR data were successfully used to improve the Radarsat-2 (extra fine mode) and Sentinel-1A/B derived PS geolocation along a railway line [21,22].

In recent years, many research studies have been focusing on hazard assessments concerned with surface deformation [23–26]. Machine learning algorithms (MLAs) were applied to evaluate landslide risks for slow mass movements, providing new insights on how to develop risk management strategies worldwide [23]. Disaster risk index methods and analytic hierarchy process (AHP) along with geographic information system (GIS) tools were used to analyze risks of land subsidence in the Kathmandu Valley, Nepal, and along the Tianjin coastal area [25,26]. However, the precision of risk analyses of surface buildings based on land subsidence is limited when fine contour data of the buildings are missing. Therefore, we propose a hazard analysis of surface buildings that are classified by LiDAR techniques.

This study focuses on the improvement of the geolocation quality of Sentinel-1A and TerraSAR-X derived PSs and on classifying structures to detect the hazard associated with urban buildings under land subsidence in Shenzhen by an integrative use of InSAR and LiDAR techniques. Section 2 describes the methods to capture the specific subsidence of buildings including the acquisition of radar scatters, the removal of invalid radar scatters, the association of radar scatters with the lidar points, the approach to classify radar scatters along the contour line of buildings, and the establishment of a hazard assessment method. The different structures of our test site, the city of Shenzhen, especially its construction sites and reclamation areas, are depicted in Section 3; Section 4 addresses the experimental results and discusses the hazard assessment system concerned with building subsidence. The conclusions follow in Section 5.

## 2. Methods

In order to monitor and identify the land subsidence of urban buildings and to assess the buildings' hazard level associated with subsidence, two basic technologies, InSAR and LiDAR, were used in combination. The workflow of the study is shown in Figure 1. It comprised obtaining subsidence information by radar scatters based on TS-InSAR technology and the removal of invalid radar scatters. Further, it comprised the geolocation improvement of radar scatters based on LiDAR technology, extracting the contour lines of buildings, and the establishment of the potential subsidence hazard assessment system of buildings.

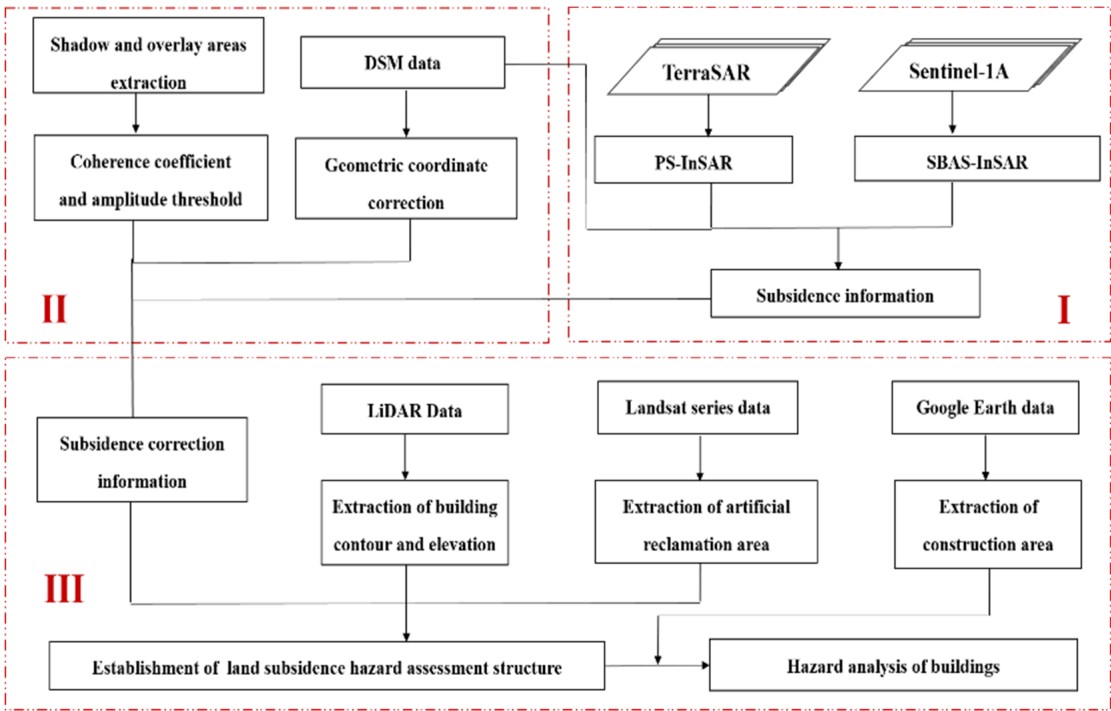

**Figure 1.** Workflow of the study.

### 2.1. TS-InSAR Technology

This section reviews the SBAS and PS-SBAS algorithms. Differential interferograms of the two methods both were formed through the selection of appropriate spatial and temporal baselines for SAR images to avoid weakening of temporal and spatial decoherence factors. If there are N scene images, M interferograms are formed as follows:

$$\frac{N+1}{2} \le M \le N\left(\frac{N+1}{2}\right) \tag{1}$$

The two methods follow different strategies to derive the target deformation information. For the SBAS-InSAR method, the phase equation of all interferograms was formed according to the subset condition of the free combination interferogram, and the least square method or singular value decomposition (SVD) method was used to estimate the deformation parameters [27,28]. In the actual processing, the temporal and spatial filtering method was used to remove the atmospheric delay phase and to separate the nonlinear deformation. The sum of the estimated low-frequency deformation and the nonlinear deformation represents the entire deformation of the study area. The phase time series of pixel x of the interferogram along the radar line of sight obtained by the small baseline method is shown in Equation (2), where *i* represents the time.

$$\varphi(t_{i,x}) = \frac{4\pi}{\lambda} d(t_{i,x})(i = 1, \dots N) \tag{2}$$

On the other hand, the deformation information of the PS-SBAS-InSAR method was obtained by using the phase change characteristics of PS points that were selected by analyzing the coherence map and intensity map of SAR interferograms [29,30]. The interference phase at the PS point of *i* in the interferogram of k can be expressed by Equation (3).

$$\hat{\varphi}_i = \varphi_{def\_i} + \varphi_{topo\_i} + \varphi_{atm\_i} + \varphi_{noise\_i} \tag{3}$$

where $\varphi_{def\_i}$ represents the deformation phase information along the line of sight, $\varphi_{topo\_i}$ represents the phase information of the elevation error introduced, $\varphi_{atm\_i}$ is expressed as

atmospheric delay phase, and $\varphi_{noise\_i}$ is other noise phases. The Delaunay triangulation network was used to construct the network, and the phase difference model of adjacent PS points was established to solve the model parameters to obtain the surface subsidence information. By solving the functional model, the additional atmospheric phase, the errors of the digital elevation model (DEM), and other noises of PS points were removed, and the accurate deformation of the land surface could be obtained [31–33].

### 2.2. Geometric Distortion Region Elimination and Geolocation Improvement by LiDAR Data

The elimination of geometric distortions introduced by SAR recordings of urban buildings or terrain and the geolocation improvement of radar scatters to avoid the dislocation of subsidence PSs are discussed in this section. The distortions of SAR imagery are manifold and depicted as shadows, layover, foreshortening, etc. [34]. Shadowing is caused by the occlusion effect of the building that has no echo information, and the intensity is extremely low in the amplitude map. The shadow areas can be removed by setting a higher coherence threshold in the TS-InSAR technology [35]. The signal intensity of the layover areas is composed of multiple scattered surface echoes, and subsequently, the echo intensity is significantly higher than that of the surrounding. A layover area can be determined by threshold segmentation of the amplitude map [36].

Besides the abovementioned removal of invalid radar scatters, a geometric precise correction of SAR images was performed in this experiment, based on accurate information of a DSM with 0.5 m high-precision data recorded by airborne LiDAR technique. During the TM-InSAR processing, PS geolocation including azimuth direction (radar signal direction) and range direction (satellite flight direction) was refined by subpixel registration operation based on DSM data. The simple coordinate correction formula is as follows:

$$r_p = r_p\prime + \Delta r_p \tag{4}$$

$$a_p = a_p\prime + \Delta a_p \tag{5}$$

where $r_p$ and $a_p$ represent the raw range and azimuth position of PS, respectively. $(r_p\prime, a_p\prime)$. $r_p$ and $a_p$ represent the refined range and azimuth position, respectively. $\Delta r_p$ is the coordinate correction value in the range direction, and $\Delta a_p$ denotes the coordinate correction value in the range direction.

With the high spatial resolution of a DSM, the accuracy of subpixel positioning can easily be improved in a two-dimensional manner. Further, the accurate DSM data were treated as (reference) real-object data, which can be used to refine PS geolocation with the error ellipsoids by computing the offset between them. The nearest neighbor linking (NNL) approach was applied to compute the offsets between feature points of the DSM and the InSAR derived PS points to preliminary realize the geometric procedure [37]. The high precision geometric correction can overcome the geometric deformations introduced by the SAR recording technique and the mismatch between real objects.

### 2.3. Buildings Extraction and Model Establishment

The final procedure conducted was a potential hazard assessment of buildings. Since the cumulative land subsidence, subsidence rate, building elevation, and the distribution of the land reclamation area are closely related to the hazard assessment of land subsidence [25], they were all selected as factors of assessment structure and classified to a single index by assigning a value. As the above four indices show, a nonlinear behavior, the sum of the score and weight product of the corresponding grades of each evaluation factor (comprehensive score) was quantitatively calculated by the comprehensive index method. Moreover, the land subsidence hazard areas were divided according to certain standards. The formula of the composite index method is as follows:

$$A_{hazard} = \sum_{i=1}^{n} (A_i \times B_i) \tag{6}$$



where $A_{hazard}$ is the comprehensive hazard index; $A_i$ represents the score of the evaluation factor; $B_i$ is the weight of the evaluation factor; and $n$ represents the number of evaluation factors.

To evaluate the hazard of buildings as related to land subsidence, it is necessary to extract the building contours of the study area. The operation was adaptively implemented based on the high-precision point cloud data obtained by LiDAR technology. The urban point cloud data contains roads, bridges, grasslands and other types of data, which need to be classified to extract all the buildings. Next, a feature combination method was applied to extract the spatial features including corner points, contour lines, and roof patches [38]. Finally, the complete contours of the buildings in the main subsidence area were obtained. Apart from the specific subsidence measured, further information for the above building hazard assessment was obtained through other remote sensing technologies. Finally, the hazard of buildings with land subsidence was analyzed.

## 3. Test Site and Data Used

The area selected for our investigations is the city of Shenzhen, which is located in the south of the Guangdong Province in China. Numerous large-scale land reclamation projects have been implemented to meet the requirements for rapid economic development and the increasing demand for urban expansion. Figure 2 shows the study area and the regions recorded for analyses by TerraSAR-X and Sentinel-1A spaceborne instruments. It further displays the significant margin between the coastline of the 30 m SRTM DEM obtained in the year 2000 (color image) and the coastline of the current district boundary map (black line) obtained in 2020, attributable to huge land reclamation projects. Such large-scale reclamation of land was not given enough time to realize the sufficient deposition of unstable alluvial clay. Thus, the soils became compacted under the pressure from high-intensity constructions and self-consolidation of the clay. The compaction process led to ground deformations, resulting in serious environmental problems such as obvious land subsidence, which, in the worst case, can lead to the inclination and collapse of buildings. These ongoing slow but dangerous processes represent a serious future danger for Shenzhen, which demands precise monitoring and careful hazard analyses of building in the future [39,40].

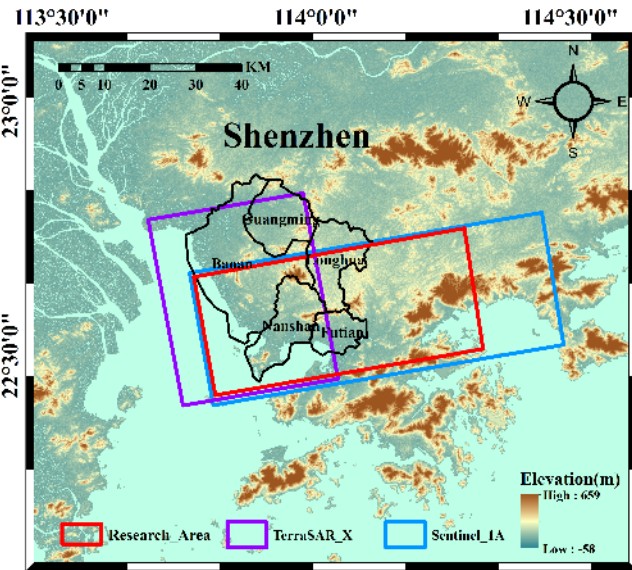

**Figure 2.** Location of the research area and frames of SAR datasets used in the study.

The Shenzhen region was covered by a total of 81 Sentinel-1A recordings from March 2017 to April 2019 by the ascending satellite track number 11, and by 10 TerraSAR-X scenes with a spatial resolution of 3 m from August 2019 to May 2020 by the ascending track

number 404. Both satellites traveled from approximately south to north at inclinations of 98.2° and 97.4°, respectively, and looked (and recorded) to the east during the ascending node. The orbit repeat cycle of the Sentinel-1A satellite was 12 days, while the repetition rate of the TerraSAR-X satellite was 11 days. An external DSM was used to remove the influence of the terrain errors and DEM errors, in addition to refining computed InSAR PS geolocations in the subsequent differential interferometry. The basic SAR parameters are given in Table 1. Tables 2 and 3 show the detailed parameters of the LiDAR and the DSM data of Shenzhen City.

**Table 1.** Basic parameters of the used SAR satellite systems.

| Sensor | TerraSAR-X | Sentinel-1A |
|---|---|---|
| Band Wavelength (cm) | X(3.1) | C(5.6) |
| Incident angle (°) | 35.28 | 34.04 |
| Slant range spacing (m) | 0.9 | 2.3 |
| Azimuth spacing (m) | 2 | 14.0 |
| Pass direction | Ascending | Ascending |
| Track number | 404 | 11 |
| Number of scenes | 10 | 81 |

**Table 2.** Basic parameters of the used LiDAR data.

| Parameters | |
|---|---|
| Scan frequency (Hz) | 344.8 |
| Average point spacing (m) | 0.46 |
| Average point density (pts/m$^2$) | 4.66 |
| Mean square error of elevation (m) | Flat/0.15, hill/0.35, mountain/0.5 |
| Horizontal datum | CGCS 2000 |
| Elevation datum | 1985 national elevation datum of China |

**Table 3.** Basic parameters of the used DSM data.

| Parameters | |
|---|---|
| Resolution (m) | 2 |
| Projection mode | Gauss Kruger projection three-degree zonation |
| Horizontal datum | CGCS 2000 |
| Elevation datum | 1985 national elevation datum of China |

## 4. Results and Discussion

Land subsidence rates in Shenzhen were calculated by applying the classical and multi-temporal InSAR techniques to the Sentinel-1A and Terra-X SAR data. The available 81 ascending Sentinel-1A images were used for calculations based on the SBAS algorithm. Two key parameters, including time baseline and space baseline, were set to 75D and 150 m, respectively. Then, 333 differential interference pairs with high quality were computed with a multilook factor of 4 × 1 in range and azimuth directions, leading to a 15 × 15 m pixel size on the ground. The minimum cost flow method was used for phase unwrapping, with the coherence threshold set to 0.2. Atmospheric errors were corrected by a polynomial model and atmospheric filtering. This method computes land subsidence time series and residual topographic heights using the SVD least-squares inversion technique. The velocity fields were obtained by using a stacking technique [41]. Figure 3 shows the vertical subsidence rates per year and the cumulative vertical subsidence rates from March 2017 till April 2020. The positive value (blue color) of velocities indicates that the surface rises in the vertical direction, and the negative value (red color) indicates that the surface sinks. It is found that the vertical line-of-sight velocities range from −40 mm to 12 mm, with the largest subsidence position located near the Bao'an Airport.

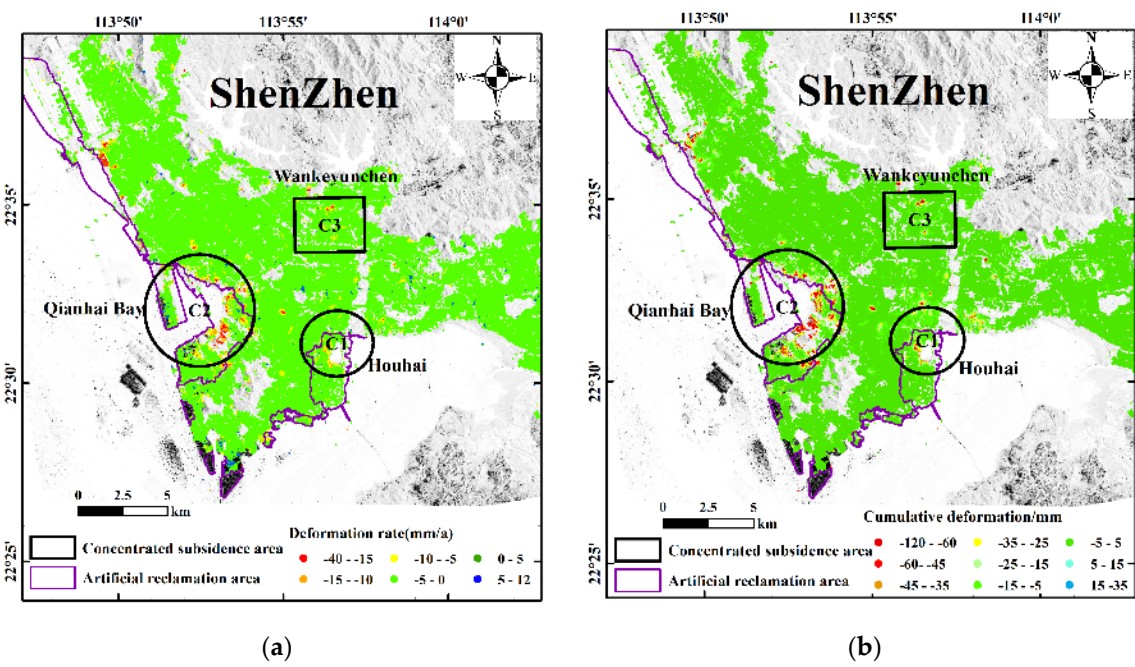

**Figure 3.** (**a**) Annual vertical subsidence rates based on Sentinel-1A images; (**b**) cumulative vertical subsidence rates from March 2017 till April 2020 based on Sentinel-1A images. C1 to C3, see text.

Besides the multitemporal analysis based on Sentinel-1 recordings, additional SAR data with a very high resolution of 3 m were exploited to investigate and validate land subsidence rates further. In total, 10 ascending TerraSAR-X images were analyzed to capture and measure subsidence rates between August 2019 and May 2020 based on the SBAS-PS-InSAR processing method. Overall, 42 interferograms were calculated with a multilook factor of 2 × 2 in range and azimuth directions, respectively, leading to a 6 × 6 m pixel size on the ground. One million highly coherent PS points were selected for the PS-InSAR process [42]. The adaptive filtering method was used to remove atmospheric errors, and the average annual vertical deformation rates were calculated. Figure 4 displays the refined vertical subsidence rates per year and the cumulative variables derived from TerraSAR-X data. The vertical subsidence rates vary from −22 mm to 20 mm, and the vertical cumulative subsidence rates vary from −17 mm to 15 mm.

As depicted by the displacement maps, most land subsidence areas are located in the artificially raised reclamation area. A most possible reason is that the artificial reclamation areas and construction sites that even might interfere, influence the rate of subsidence. Subsidence velocities in the Shenzhen District vary spatially based on two datasets, which also suggests that the urban area is stable, except for the obvious subsidence trends observable in the C1 to C3 subregions. The C1 zone is located in Houhai, which has several small land subsidence funnels nearby. The vertical subsidence velocities of the C1 zone vary from −21.4 mm to 6.5 mm based on the Sentinel-1A images, while the subsidence velocities vary from −19.9 mm to 17.0 mm based on the TerraSAR-X images. Zone C2 is located in Qianhai Bay, which is the largest area of artificial reclamation and has developed recently into a huge commercial trade center of the city with many high-rise buildings. The vertical subsidence velocities of the C2 zone vary from −28.3 mm to 10 mm for Sentinel-1A images, and from −22 mm to 9.5 mm for TerraSAR-X images. The C3 zone located in the center of Shenzhen near Wankeyunchen comprises areas with fewer deformations than those occurring in the subregions C1 and C2. The vertical subsidence velocities of the C3 zone vary from −27.5 mm to 2.6 mm for Sentinel-1A images, and from −18.4 mm to 5.8 mm for TerraSAR-X images.

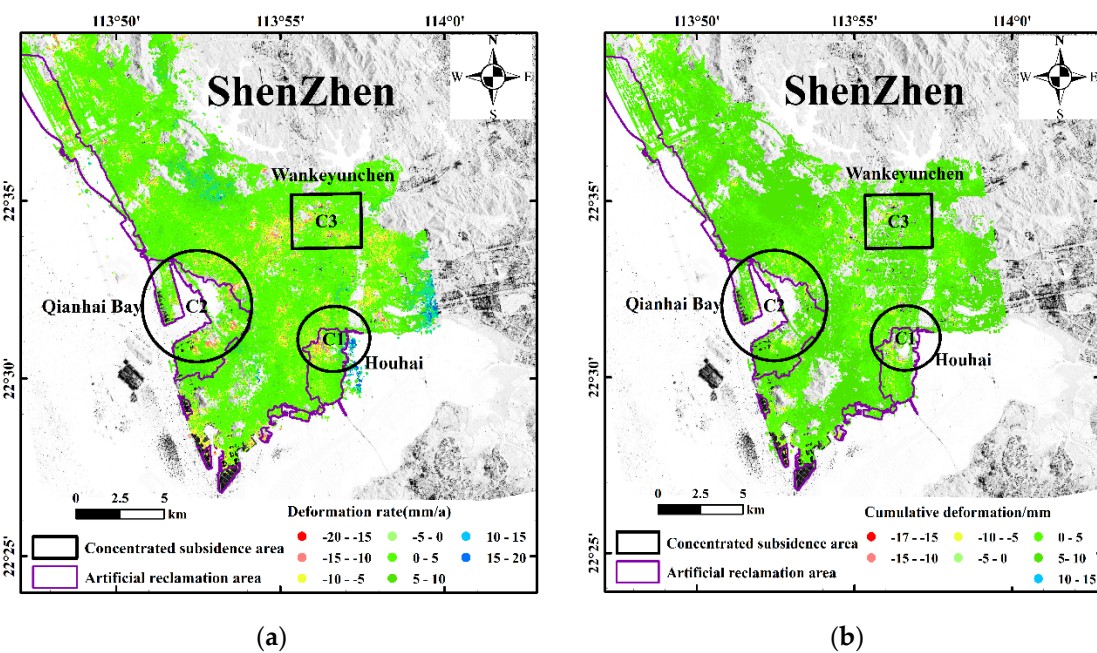

**Figure 4.** (**a**) Annual vertical subsidence rates based on TerraSAR-X images; (**b**) cumulative vertical subsidence rates based on TerraSAR-X images from August 2019 to May 2020. C1–C3, see next.

In recent years, many scholars have monitored the land subsidence of Shenzhen City in different time periods based on SAR data. In 2018, Liu Peng et al. found that the main subsidence areas calculated by multisensor SAR data are located in the coastal areas of Shenzhen, including Qianhai, Houhai, and the area south of the Bao'an International Airport. In 2013, Hu Zheng et al. found land subsidence rates with an increasing trend in the Qianhai Bay area of Shenzhen based on ENVISAT SAR data recorded between 2007 and 2010 using SBAS InSAR technology. The cumulative subsidence in local areas reached 60 mm, with average annual subsidence of 22 mm. In 2016, based on multisensor SAR data, Xu Bing et al. found significant subsidence rates of up to 25 mm per year in line-of-sight (LOS) direction. These occurred in the artificial reclamation areas including the Shenzhen Airport, the Bao'an Center, the Qianhai Bay, and the Shenzhen Bay. He predicted that the deformation would continue in the near future. It is found that the main subsidence areas depicted by the above InSAR studies are basically consistent with the results of this paper. However, due to different data acquisition periods and data characteristics, the specific annual average subsidence rate is slightly different.

In order to evaluate the internal coincidence accuracy of the annual subsidence rates obtained from Sentinel-1A and TerraSAR-X imagery, the standard deviation (STD) was statistically analyzed. Figure 5 shows the STD distribution of the annual subsidence rates, obtained by calculating the linear fitting deviation. If a PS point shows a strong nonlinear trend, it will produce a large deviation from the corresponding linear model. It can be seen that the standard deviation of the subsidence rate of Sentinel-1A is within 3 mm per year, and the standard deviation of the subsidence rate of more than 98.7% of PS points is less than 1 mm per year. Moreover, the standard deviation of the subsidence rate of TerraSAR-X data is within 5 mm per year, and the standard deviation of the subsidence rate of more than 86.2% of PS points is less than 1 mm per year. Thus, SBAS-InSAR technology and PS-SBAS-InSAR technology both produced reliable results.

To reduce the influence of geometric distortions expressed by shadowing, layover, and perspective in SAR imagery, the following correction methods were applied. To eliminate shadows, the coherence coefficients of PS points in the PS-SBAS-InSAR methodology were high enough. However, using the SBAS-InSAR methodology, higher coherence thresholds needed to be set. To compensate for the layover caused primarily by high-rise buildings, a threshold segmentation of the SAR amplitude map was conducted. Then,

the actual ground objects were exploited as a reference to calibrate and refine the layover area of the main subsidence site. Figure 6a,c shows local SAR images of Qianhai Bay where the layover phenomenon introduced by high buildings is clearly visible, with emphasis on the TerraSAR-X scene. Figure 6b,d displays the layover areas of Sentinel-1A (green) and TerraSAR-X (magenta) SAR images extracted by threshold segmentation of the amplitude map.

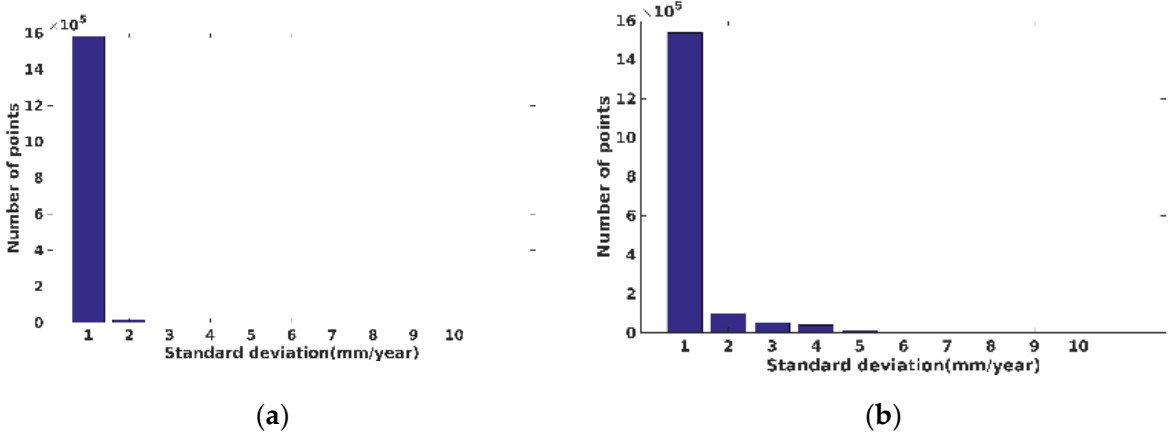

(**a**)　　　　　　　　　　　　　　　　　(**b**)

**Figure 5.** Distribution of the standard deviation of land subsidence rates: (**a**) Sentinel-1A recordings; (**b**) TerraSAR-X recordings.

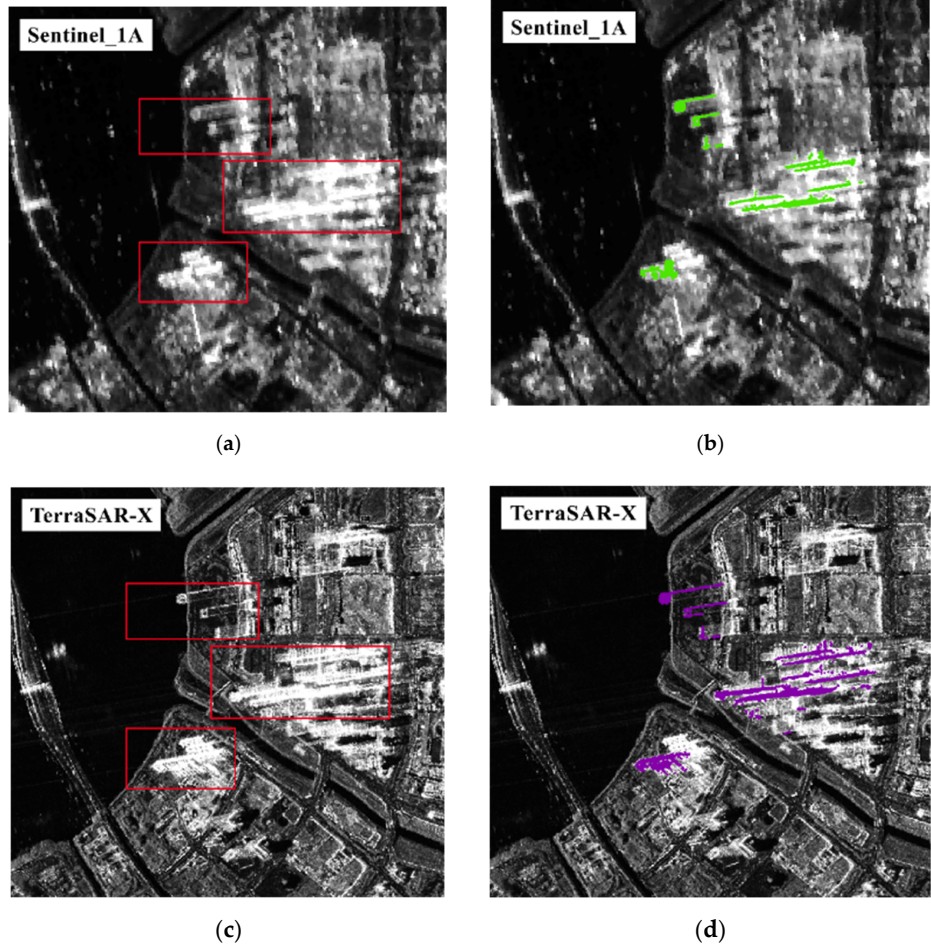

(**a**)　　　　　　　　　　　　　　　　　(**b**)

(**c**)　　　　　　　　　　　　　　　　　(**d**)

**Figure 6.** (**a**) Sentinel-1A image of the Qianhai Bay; (**b**) layover areas in the Sentinel-1A image; (**c**) TerraSAR-X image of the Qianhai Bay; (**d**) layover areas in the TerraSAR-X image.

PS points corresponding to buildings are geometrically often seriously distorted, and there is a certain deviation when superimposed on an urban laser point cloud model, as shown in Figure 7a. The red and green dots represent the PS points of the Sentinel-1A and TerraSAR-X data, which display a distinct geometric deviation to buildings. The lower the resolution of the SAR images is, the more serious the deviation. To obtain an improved match between the SAR PS points and the urban laser point cloud model, the SAR points need to be refined.

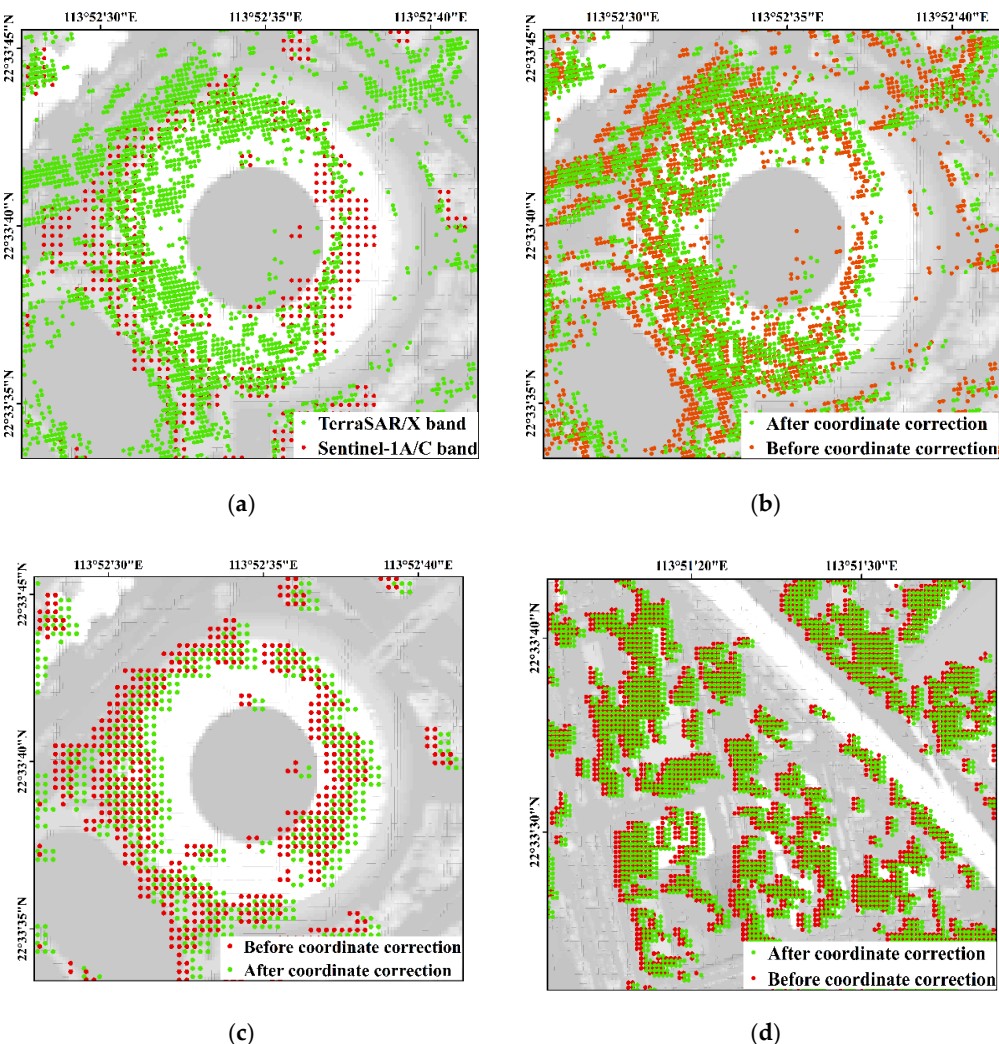

**Figure 7.** (**a**) Superposition effect of SAR PS points based on two bands and DSM data. Red and green represent the PS points based on Sentinel-1A and TerraSAR-X images, respectively; (**b**) geometric correction of TerraSAR-X subsidence for buildings; (**c**,**d**) geometric correction of Sentinel-1A subsidence for buildings.

The DSM covering Shenzhen with a grid of 2 m × 2 m derived by airborne LiDAR recordings was taken as control information to realize the high-precision geometric correction. Figure 7b shows the InSAR match before and after geometric correction based on TerraSAR-X images, while Figure 7c,d illustrates the InSAR match before and after geometric correction based on Sentinel-1A images. The deviation of originally 4–5 pixels (more pixels for building) is reduced to about 2–3 pixels in the above operation, while the deviation of originally 3–4 pixels is reduced to about 1–2 pixels. The detailed variations of PS points during the geometric correction are shown in Table 4. Sentinel-1A-derived PS points of buildings increase by 1210, while TerraSAR-X-derived PS points of buildings decrease by 13191. The inconsistency in the change of control points can be related to the fact

that the PS points of Sentinel-1A data obtained by SBAS-InSAR technology mainly include buildings, while the high-resolution TerraSAR-X data obtained by PS-SBAS-InSAR technology additionally include roads and bridges. Therefore, after geometric correction, the number of buildings PS points based on low-resolution Sentinel-1A data increases, while the number of buildings PS points based on high-resolution TerraSAR-X data decreases.

**Table 4.** Statistics of geometric correction results of building PS points.

|  | TerraSAR-X/PS Points | Sentinel-1A/PS Points |
|---|---|---|
| Before coordinate correction | 205,898 | 83,526 |
| After coordinate correction | 192,707 | 84,736 |

For a detailed analysis of the distribution of dislocation rates in the study area, the contours and heights of buildings in Shenzhen have to be determined. However, the ground resolution of the InSAR data used is less suitable for this purpose due to a ground resolution of only 6 to 15 m. Thus, based on an urban laser point cloud model derived from aircraft operations, a feature combination method was applied to map the contours of all buildings. In parallel, the altitudes of the center points of all contours were calculated to obtain the heights of the respective buildings, whereby the accuracy is ±5 m. Figure 8a shows the contours and the heights of the buildings in the three main subsidence areas C1–C3.

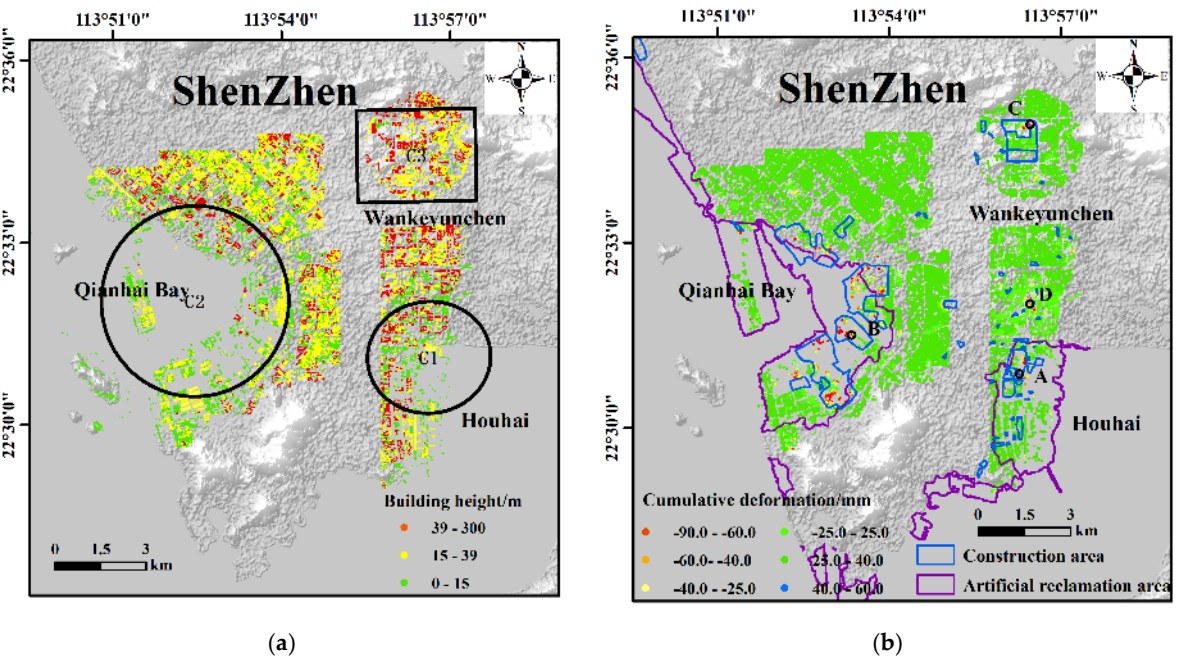

(**a**)            (**b**)

**Figure 8.** (**a**) Contour and height of buildings based on LiDAR data; (**b**) artificial reclamation area, construction area and vertical deformation rate of buildings from March 2017 to April 2020 based on Sentinel-1A images. Locations of time series data: A = Ali Center, B = Qianwan law building, C = Vanke Cloud city, D = and Yuehaimen zone.

The artificial areal accretion of land along the coast in the timeframe of 2000 to 2020 is illustrated in Figure 8b. It was mapped using a Landsat-7 ETM+ image acquired in 2000 and a Landsat-8 OLI image recorded in 2020. The changes of the coastline in the study area were mapped [43] and the water areas were classified. The construction areas were delineated by visual interpretation of multitemporal Google Calendar optical images. According to the refined subsidence rates, assisted by Google map's surface building environment information, it is found that the strongest subsidence rates in the study area are associated with construction sites during different time periods and degrees. To reveal more details

on this subject, the construction subareas Houhai, Qianhai Bay, and Wankeyunchen were mapped from March 2017 to April 2020 using Google maps with a high resolution of 0.5 m (Figure 8b).

Time series of four specific locations within the subareas were selected and the respective subsidence rates were calculated based on TerraSAR-X and Sentinel-1A data (Figure 9.) Point A is located in the Ali Center near Houhai Dengliang. The center subsided slowly in the early stage while rapidly sinking in the later stage. Point B is located near the Qianwan law building, and its changes over time are smooth and linear. There is a large number of deformations near this target point, and it is verified that the area has been leveled during the monitoring period. Point C is located near Vanke Cloud city, and the target point of this position was subsiding with time in the early stage. Point D is located near the Yuehaimen zone, and the target point of this position is subsiding with time. Subway line 12 is being built near this point, and the subsidence is surely caused by construction activities in the underground. The subsidence trend of the two types of SAR data over time is basically comparable. The verification through information about ground consistency and construction schedules unveil the encountered subsidence as primarily caused by the existing subsurface conditions and the weight of high-rise buildings. Furthermore, the high rates of subsidence mainly occur during the time of construction. In the future, the local authorities may consider a longer necessary timeframe for natural soil compaction in artificial reclamation areas or instigate the utilization of available human-made techniques to densify the underground further.

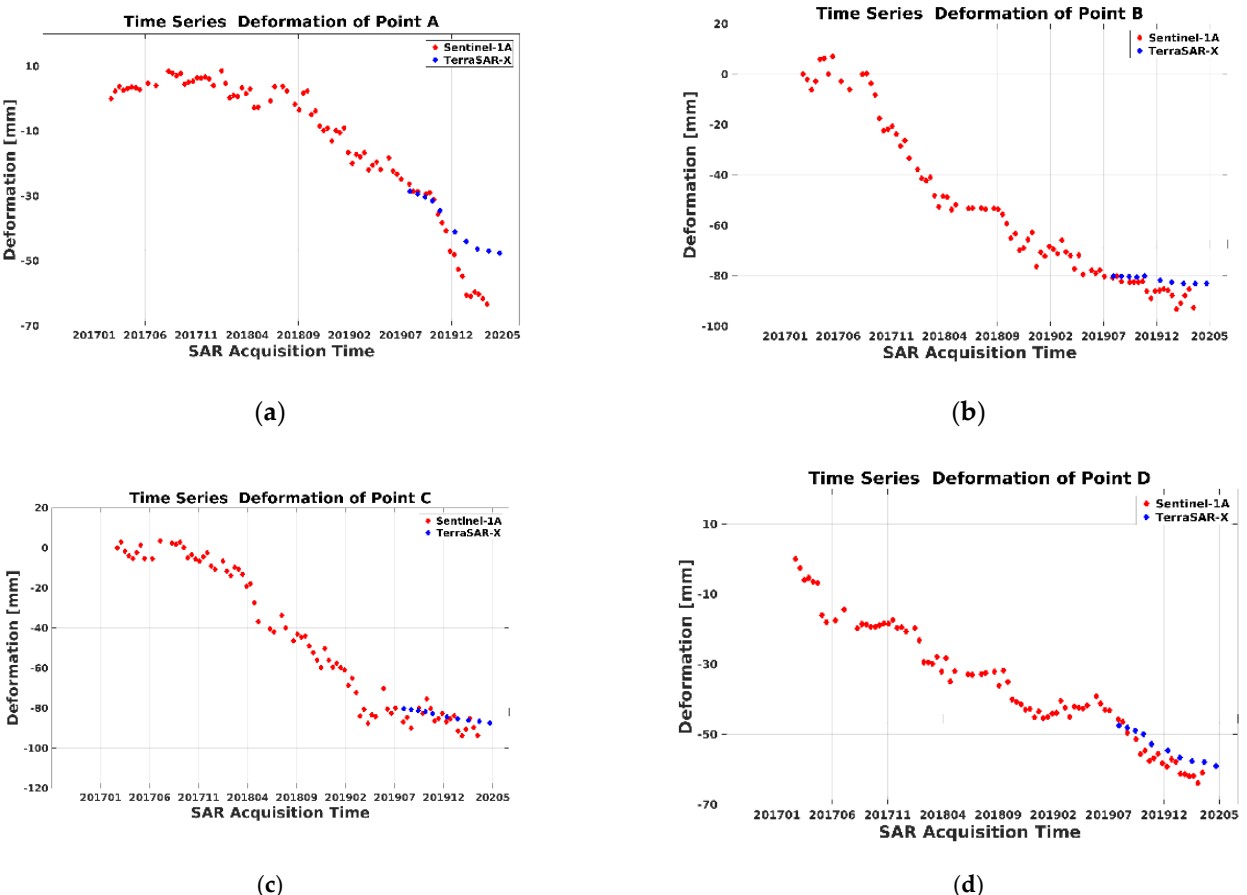

**Figure 9.** Detailed cumulative time series of land subsidence rates obtained from Sentinel-1A (red diamonds) and TerraSAR-X (blue diamonds) data. The investigated four main subsidence centers are located in (**a**) the Ali Center (A), (**b**) the Qianwan law building (B), (**c**) Vanke Cloud city (C), and (**d**) Yuehaimen zone (D). See also Figure 8.

To evaluate the building hazard in the main subsidence area, a reasonable hazard assessment of land subsidence was applied based on the comprehensive index method. This comprises four factors (the cumulative land subsidence, the land subsidence rate, the building elevation, and the distribution of land reclamation area) and three second-level indicators. The cumulative land subsidence and land subsidence rate factor are the key indicators in evaluating the land subsidence hazard because land subsidence is a slow-onset geohazard. According to the different time domains and spatial resolutions of the two data sets used, Sentinel-1A- and TerraSAR-X-derived data were set as the second indicators of the cumulative land subsidence and land subsidence rate factor. Generally, the building height is an important hazard factor. The natural settlement time of the soil in the artificial reclamation area is less, and the weight of the building itself makes the buildings in this area generally unstable. Based on indicators in determining the land subsidence hazard, all of them were weighed in Table 5. Every indicator was classified into three clusters except the distribution of the land reclamation area factor. Each datum with the former three factors was assigned a ranking number ranging from 1 to 3, in which three indicates the highest risk.

**Table 5.** Factors for indices to assess land subsidence hazard.

| Factor | Weight Value | Indicator | Weight Value | Grade Value | | |
|---|---|---|---|---|---|---|
| | | | | Low (1) | Medium (2) | High (3) |
| Accumulated land subsidence (mm) | 0.393 | Sentinel-1A (201703–202003) | 0.6 | 0–25 | 25~70 | over 70 |
| | | TerraSAR-X (201908–202005) | 0.4 | 0–4.5 | 4.5~15 | over 15 |
| Land subsidence rate (mm/a) | 0.311 | Sentinel-1A | 0.6 | 0–6 | 6~18 | over 18 |
| | | TerraSAR-X | 0.4 | 0–6 | 6~18 | over 18 |
| Building elevation (m) | 0.126 | data | data | 0–10 | 10~39 | over 39 |
| The distribution of land reclamation area | 0.170 | | | **Low(0)** | | **High(1)** |
| | | | | No land reclamation area | | Land reclamation area |

Considering the relationship between the accumulated subsidence and the annual subsidence rate indicated by each SAR record and their impact on the hazard level, the weights are set to 0.393 and 0.311, respectively. In order to reflect the degree of land subsidence, the annual rate was selected as the hazard index, and each index was divided into three equal parts with values of 1, 2, and 3, respectively. Considering SAR imaging characteristics and the time domains, the weights of Sentinel-1A and TerraSAR-X derived data were set to 0.6 and 0.4, respectively. The weight for the elevation of buildings is determined as 0.126, and the index is further divided into three equal parts with values of 1, 2 and 3, respectively. Finally, considering the impact of the distribution of the land reclamation area factor on subsidence hazard, the weight was set to be 0.170, and the index was divided into two equal parts, namely, artificial reclamation area and nonartificial reclamation area with values of 0 and 1, respectively.

The contours and heights of buildings in the major subsidence areas were automatically extracted based on the airborne LiDAR data (Figure 8a). The subsidence hazard level for the respective buildings was evaluated by Equation (6) according to the land subsidence hazard assessment system. Proportionally to the calculating weighted value, the building risk level was divided into three levels including low (less than or equal to 1.2), medium (greater than 1.2 and less than or equal 1.6), and high levels (greater than 1.6). A color code indicates three levels of risk associated with individual subsidence for each building in the area (Figure 10). It is found that some medium and high-hazard buildings display in major subsidence areas in Figure 10, especially in the artificial reclamation area of Qianhai Bay and Houhai and less in the Vanke Cloud city. Corresponding to the construction area

for calculation, the risks of individual buildings and building blocks generally increase, especially for those located closer to the coastline. This seems logical in some way, as the consolidation of the ground might decrease with the age of the reclaimed land and thus with the distance to the seashore. Furthermore, it is found that the high-rise buildings were built continuously during 2017–2020 in Qianhai Bay and Houhai. The medium and high-risk individual buildings are almost all located in the construction area marked with a magenta color line. This is basically consistent with the results of the assessment system where the urban construction activities dominate the land subsidence of urban buildings in Shenzhen. Slightly different from the results of other researchers, hazard buildings are mainly caused by building construction operation but less affected by the natural subsidence of artificial reclamation area and the subsidence of building weight, which also shows that the government and part of the government are wise in decision making. Moreover, it must be noted that different techniques of construction may additionally influence the subsidence levels of high-rise buildings; therefore, their dislocations and safety should be strictly controlled during the whole construction process and the subsequent years till their final consolidation. Of course, it is worth noting that land subsidence in Shenzhen has existed for a long time; thus, continuous monitoring of building subsidence is also very important.

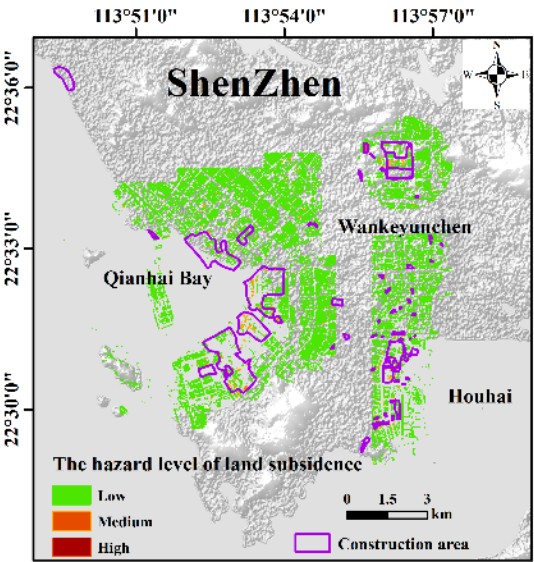

**Figure 10.** Hazard assessment of building in three main subsidence areas.

## 5. Conclusions

Land subsidence of buildings in Shenzhen was calculated and analyzed by means of advanced InSAR and LiDAR techniques. Further, hazard levels associated with land subsidence of high-rise buildings were assessed by an evaluation system. By the use of the Sentinel-1A time series from the year 2017 to 2020, land subsidence rates between −40 mm/year and 12 mm/year could be derived. The ground structure proved to be almost stable in the middle and northeast of Shenzhen except the main subsidence areas in Houhai, the Qianhai Bay, and an area near Wankeyuncheng. Land subsidence rates calculated by the use of TerraSAR-X time series (2019–2020) show comparable patterns to the results derived from Sentinel-1A images. When the contour lines of buildings are extracted on the basis of LiDAR technology and the coordinates of SAR imaging distortion points are refined by LiDAR as well, it is found that only minor buildings show ground subsidence in the three subregions C1–C3. To refine the subsidence risk of these buildings further, a land subsidence hazard assessment system was established. It is based on the annual average subsidence rate, the cumulative subsidence of InSAR, the distribution of the artificial reclamation area, and the height of the buildings. It is found that some buildings

with medium and high-grade risk are located in Qianhai Bay and Houhai. According to the available historical construction information derived from Google maps analyses, most buildings are assigned to be at high hazard during their construction phases or by underground activities such as the subway construction in Houhai.

On the one hand, the subsidence information of some buildings may be missed due to the constraints associated with the SAR geometric recording technique. On the other hand, the safety of each building mainly depends on the subsidence of its construction materials in different directions. We further like to state that the strategy and algorithms developed and applied are transferable to other sites concerned, as far as the relevant InSAR and LiDAR data are available. In the future, our work will concentrate on the application of multisource high-resolution SAR imagery of different orbits to monitor the stability of buildings on a large scale and to improve the hazard assessment of infrastructures.

**Author Contributions:** All authors participated in editing and reviewing the manuscript. Y.H. implemented the methodology, analyzed the InSAR and LiDAR data, produced the results, and wrote the original paper. H.M. and J.W. analyzed LiDAR data and implemented related experiments. G.X. and H.K. supervised the research and revised the manuscript. T.L. proofread and revised the manuscript. All authors have read and agreed to the published version of the manuscript.

**Funding:** This work is supported by the Shenzhen Science and Technology Program (Grant No. KQTD20180410161218820) and the National Key Research and Development (R&D) Project (Grant No. 2016YFC0800105-01).

**Data Availability Statement:** Authors are grateful to the European Space Agency (ESA) for providing the Sentinel-1A SAR data and the precise orbit information free of charge. We also like to thank the German Aerospace Center (DLR) for the supply of the TerraSAR-X data.

**Acknowledgments:** The Lidar data were copyrighted by the Shenzhen Municipal Planning, and Land Realestate Information Center (Shenzhen GeoSpatial Information Center). The LiDAR technology was implemented by the Shenzhen Lijian Skyeye-laser Technology Limited Company. In the end, the authors like to thank the anonymous reviewers for their efforts and constructive comments to improve the quality of this paper.

**Conflicts of Interest:** The authors declare no conflict of interest.

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
