# Peer review of "Integration of InSAR and LiDAR Technologies for a Detailed Urban Subsidence and Hazard Assessment in Shenzhen, China"

_remotesensing, doi:10.3390/rs13122366_

Round 1
Reviewer 1 Report
First, I would like to thanks the authors for their interesting article. The fusion of microwave and LiDAR data is not common practice. Although the presented topic is interesting I have some issues concerning the content.
First, please correct the images: Figures 3- 5,7,9, the texts in the images are not clear enough. Next, there are two Figures 9 (line 320 and line 371).
The presented methodology is correct. Data supplementation and object classification based on LiDAR data are correct, and if such data is available, it should be used. However, my concern is about the accuracy of the results. First, about the accuracy of data co-registration, especially when using data with different resolutions. Could you provide information about the accuracy verification process? Second, the accuracy of land subsidence level determined from SAR data. Did the authors possibly access ground (in-situ) measurements of such deformation for the analysed area?
Author Response
Thank you so much! And concrete answers in following documents. Please see the attachment.

Reviewer 2 Report
Dear Authors, I have carefully read your paper "Integration of InSAR and LiDAR technologies for a detailed urban subsidence and hazard assessment in China". The study realized is of great interest. I find the paper well written and structured. However, in my opinion, the manuscript could be published after some necessary revisions, which useful to improve the paper:
The InSAR technology is widely used in the different research field. In my opinion, in Introduction section, you should provide further information about InSAR application, for example, landslide monitoring, infrastructure monitoring and landslide susceptibility. To this, I suggest adding to the literature review some more articles recently published and dealing with important aspects:
- Novellino, A., et al., (2021). Slow-moving landslide risk assessment combining Machine Learning and InSAR techniques. CATENA, 203, 105317.
- Hu, F., et al., (2019). Monitoring deformation along railway systems combining multi-temporal InSAR and LiDAR data. Remote sensing, 11(19), 2298.
- Miele, P., et al., (2021). Landslide Awareness System (LAwS) to Increase the Resilience and Safety of Transport Infrastructure: The Case Study of Pan-American Highway (Cuenca–Ecuador). Remote Sensing, 13(8), 1564.
How did you conduct the segmentation process?
What did you use?
What software did you employ?
Figure 9 has to be added the selected sub-areas on the map.
In the "conclusions" paragraph, it could be interesting to expand some considerations on the exportability of the method also in other contexts and therefore not to make the work too site-specific.
Reviewer 3 Report
The authors proposed a very interesting topic trying to monitoring ground deformations (subsidence and uplift) through InSAR tecnique.
The work was thoroughly written, the authors carried out detailed analyzes, the results are
interesting as well as the subject of the topic under consideration.
Some minor issues have to be addressed before publication, namely:
-Introduction
The authors should improve the methodology background by including a more detailed description of previous specific studies, for example:
-Nitti D.O., Nutricato R., Lorusso R.; Lombardi N, Bovenga F., Bruno M.F., Chiaradia M.T., Milillo G. (2015). On the geolocation accuracy of COSMO-SkyMed products. In: Proc. SPIE 9642, SAR Image Analysis, Modeling, and Techniques XV, 96420D
-Bruno M.F., Molfetta M.G., Mossa M., Morea A., Chiaradia M.T., Nutricato R., Nitti D.O., Guerriero L., Coletta A. (2016). Integration of multitemporal SAR/InSAR techniques and NWM for coastal structures monitoring: Outline of the software system
and of an operational service with COSMO-SkyMed data. In: Proc. 2016 IEEE Workshop on Environmental, Energy, and Structural Monitoring Systems (EESMS);
Discussion
-Can you provide a summary table where results of the geometric correction are shown?
- Can you compare your results with other studies?
For example:
Xu, B.; Feng, G.; Li, Z.; Wang, Q.; Wang, C.; Xie, R. Coastal subsidence monitoring associated with land reclamation using the point target based SBAS-InSAR method: A case study of Shenzhen, China. Remote Sens. 2016, 8, 652.
Sun, Q., Jiang, L., Jiang, M., Lin, H., Ma, P., & Wang, H. (2018). Monitoring coastal reclamation subsidence in Hong Kong with distributed scatterer interferometry. Remote Sensing, 10(11), 1738.
Moreover:
-Table 1 Please add SAR image resolution
-Figure 5. Please use a single graph when comparing results.
-Lines 389-393 must be deleted
-Add a table with Basic parameters of used data (LIDAR, DSM, etc)
Author Response
Response to Reviewer 3 Comments
Point 1: Introduction-The authors should improve the methodology background by including a more detailed description of previous specific studies.
Response 1: Thank you, we have improved the methodology background by including a more detailed description of previous specific studies about risk assessment of surface deformation.
Point 2: Discussion - Can you provide a summary table where results of the geometric correction are shown?
Response 2: Text added.
Yes, the results of geometric correction are shown in Table 4. 1210 Sentinel-1A drived PS points of buildings are increased and above 13190 TerraSAR-X drived PS points of buildings are decreased after the geometric correction. Number change of building PS points inconsistency is due to the fact that the PS points of Sentinel-1A data obtained by SBAS-InSAR technology mainly include buildings, while the high-resolution TerraSAR-X data obtained by PS-SBAS-InSAR technology includes not only buildings, but also roads and bridges. Therefore, after geometric correction, the number of buildings PS points based on Sentinel-1A data with low resolution increases, while the number of buildings PS points based on TerraSAR data with high resolution decreases. Of course, the most obvious is the change map of InSAR PS points for linear infrastructure such as roads before and after geometric correction. Due to this experiment mainly describes buildings, there is no else map added. Later, there will be related paper to introduce the geometric correction process in detail.
Point 3: Discussion - Can you compare your results with other studies?
Response 3: Text added
Yes. In recent years, many scholars have monitored the land subsidence of Shenzhen City in different time periods based on SAR data. In 2018, Liu Peng, et al. found that the main subsidence areas based on multi-sensor SAR data are located in the coastal areas of Shenzhen, including Qianhai, Houhai and the area south of the Bao'an International Airport. In 2013, Hu Zheng et al. found an increasing trend of land subsidences occurring in the Qianhai Bay area of Shenzhen based on ENVISAT SAR data (2007-2010) by SBAS InSAR technology. The cumulative subsidence in local areas reached 60 mm, with an average annual subsidence of 22 mm. In 2016, based on multi-sensor SAR data, Xu Bing et al. found that significant subsidence of up to 25 mm per year in "line of sight" direction existed in artificial reclamation areas. This includes the Shenzhen airport, the Bao'an center, the Qianhai Bay and the Shenzhen Bay. He predicted that the deformation would continue in the near future. It was found that the results calculated for the main subsidence areas by InSAR are basically consistent with the results of this paper. However, due to different data acquisition periods and data characteristics, the specific annual average subsidence rate is slightly different.
Point 4: Table 1. Please add SAR image resolution
Response 4: Yes, the resolution has been listed in the SAR parameter table 1 in slant range spacing (m) and azimuth spacing (m).
Point 5: Figure 5. Please use a single graph when comparing results.
Response 5: Because Figure 5 shows the inner accuracy of InSAR results from Sentinel and TerraSAR-X, we where compelled using two graphs.
Point 6: -Lines 389-393 must be deleted
Response 6: Yes, thank U. Text added
It is found that some medium and high-hazard buildings display in major subsidence areas in Figure 10, especially in the artificial reclamation area of Qianhai Bay and Houhai and less in the Vanke Cloud city.
Slightly different from the results of other researchers, hazard buildings are mainly caused by building construction operation, but less affected by the natural subsidence of artificial reclamation area and the subsidence of building weight, which also shows that the government and part of the government are wise in decision-making
Of course, it is worth noting that land subsidence in Shenzhen has been exists, thus continuous monitoring of building subsidence is also very important.
Point 7: Add a table with Basic parameters of used data (LIDAR, DSM, etc)?
Response 7: Yes, it is added in table 2 and 3 now.